# CTRL: Closed-Loop Transcription to an LDR via Minimaxing Rate Reduction

**DOI:** 10.3390/e24040456

**Published:** 2022-03-25

**Authors:** Xili Dai, Shengbang Tong, Mingyang Li, Ziyang Wu, Michael Psenka, Kwan Ho Ryan Chan, Pengyuan Zhai, Yaodong Yu, Xiaojun Yuan, Heung-Yeung Shum, Yi Ma

**Affiliations:** 1Department of EECS, University of California Berkeley, Berkeley, CA 94720, USA; daixili_cs@163.com (X.D.); tsb@berkeley.edu (S.T.); psenka@berkeley.edu (M.P.); yaodong_yu@berkeley.edu (Y.Y.); 2School of Computer Science and Engineering, University of Electronic Science and Technology of China, Chengdu 610056, China; xjyuan@uestc.edu.cn; 3Tsinghua-Berkeley Shenzhen Institute, Shenzhen 518055, China; lmy17@mails.tsinghua.edu.cn; 4International Digital Economy Academy, Shenzhen 518048, China; robinwuzy@gmail.com (Z.W.); msraharry@hotmail.com (H.-Y.S.); 5Mathematical Institute for Data Science, Johns Hopkins University, Baltimore, MD 21218, USA; kchan49@jhu.edu; 6Institute for Applied Computational Science, Harvard University, Cambridge, MA 02138, USA; pzhai@g.harvard.edu

**Keywords:** closed-loop transcription, linear discriminative representation, rate reduction, minimax game

## Abstract

This work proposes a new computational framework for learning a structured generative model for real-world datasets. In particular, we propose to learn *a **C**losed-loop **Tr**anscription*between a multi-class, multi-dimensional data distribution and a ***L**inear discriminative representation* (*CTRL*) in the feature space that consists of multiple independent multi-dimensional linear subspaces. In particular, we argue that the optimal encoding and decoding mappings sought can be formulated as a *two-player minimax game between the encoder and decoder*for the learned representation. A natural utility function for this game is the so-called *rate reduction*, a simple information-theoretic measure for distances between mixtures of subspace-like Gaussians in the feature space. Our formulation draws inspiration from closed-loop error feedback from control systems and avoids expensive evaluating and minimizing of approximated distances between arbitrary distributions in either the data space or the feature space. To a large extent, this new formulation unifies the concepts and benefits of Auto-Encoding and GAN and naturally extends them to the settings of learning a *both discriminative and generative* representation for multi-class and multi-dimensional real-world data. Our extensive experiments on many benchmark imagery datasets demonstrate tremendous potential of this new closed-loop formulation: under fair comparison, visual quality of the learned decoder and classification performance of the encoder is competitive and arguably better than existing methods based on GAN, VAE, or a combination of both. Unlike existing generative models, the so-learned features of the multiple classes are structured instead of hidden: different classes are explicitly mapped onto corresponding *independent principal subspaces* in the feature space, and diverse visual attributes within each class are modeled by the *independent principal components* within each subspace.

## 1. Introduction

One of the most fundamental tasks in modern data science and machine learning is to learn and model complex distributions (or structures) of real-world data, such as images or texts, from a set of observed samples. By “to learn and model”, one typically means that we want to establish a (parametric) mapping between the distribution of the real data, say $x\in R^{D}$, and a more compact random variable, say $z\in R^{d}$:(1)$$f\left(\cdot ,\theta \right):x\in R^{D}\mapsto z\in R^{d}\;\mathrm{or}\;\mathrm{the}\;\mathrm{inverse}\;g\left(\cdot ,\eta \right):z\in R^{d}\mapsto x\in R^{D},$$
where z has a certain standard structure or distribution (e.g., normal distributions). The so-learned representation or feature z would be much easier to use for either generative (e.g., decoding or replaying) or discriminative (e.g., classification) purposes, or both.

**Data embedding versus data transcription.***Be aware* that the support of the distribution of x (and that of z) is typically *extremely low-dimensional* compared to that of the ambient space (for instance, the well-known CIFAR-10 datasets consist of RGB images with a resolution of 32×32. Despite the images being in a space of $R^{3072}$, our experiments will show that the intrinsic dimension of each class is less than a dozen, even after they are mapped into a feature space of $R^{128}$) hence the above mapping(s) may not be uniquely defined based on the support in the space $R^{D}$ (or $R^{d})$. In addition, the data x may contain multiple components (e.g., modes, classes), and the intrinsic dimensions of these components are not necessarily the same. Hence, without loss of generality, we may assume the data x to be distributed over a union of low-dimensional nonlinear submanifolds $\cup_{j=1}^{k}M_{j}\subset R^{D}$, where each submanifold $M_{j}$ is of dimension $d_{j}\ll D$. Regardless, we hope the learned mappings *f* and *g* are (locally dimension-preserving) *embedding* maps [1], when restricted to each of the components $M_{j}$. In general, the dimension of the feature space *d* needs to be significantly higher than all of these intrinsic dimensions of the data: $d>d_{j}$. In fact, it should preferably be higher than the sum of all the intrinsic dimensions: $d\geq d_{1}+\cdots +d_{k}$, since we normally expect that the features of different components/classes can be made fully independent or orthogonal in $R^{d}$. Hence, without any explicit control of the mapping process, the actual features associated with images of the data under the embedding could still lie on some arbitrary nonlinear low-dimensional submanifolds inside the feature space $R^{d}$. The distribution of the learned features remains “latent” or “hidden” in the feature space.

So, for features of the learned mappings (Equation 1) to be truly convenient to use for purposes such as data classification and generation, the goals of learning such mappings should not only simply reduce the dimension of the data x from *D* to *d* but also determine explicitly and precisely how the mapped feature z=f(x) is distributed within the feature space $R^{d}$, in terms of both its support and density. Moreover, we want to establish an explicit map g(·) from this distribution of feature z back to the data space such that the distribution of its image $\hat{x}=g\left(z\right)$ (closely) matches that of x. To differentiate from finding arbitrary feature embeddings (as most existing methods do), we call embeddings of data onto an explicit family of models (structures or distributions) in the feature space as *data transcription*.

**Paper Outline.** This work is to show how such transcription can be achieved for real-world visual data with one important family of models: the linear discriminative representation (LDR) introduced by [2]. Before we formally introduce our approach in Section 2, for the remainder of this section, we first discuss two existing approaches, namely autoencoding and GAN, that are closely related to ours. As these approaches are rather popular and known to the readers, we will mainly point out some of their main conceptual and practical limitations that have motivated this work. Although our objective and framework will be mathematically formulated, the main purpose of this work is to verify the effectiveness of this new approach empirically through extensive experimentation, organized and presented in Section 3 and Appendix A. Our work presents compelling evidence that the closed-loop data transcription problem and our rate-reduction-based formulation deserve serious attention from the information-theoretical and mathematical communities. This has raised many exciting and open theoretical problems or hypotheses about learning, representing, and generating distributions or manifolds of high-dimensional real-world data. We discuss some open problems in Section 4 and new directions in Section 5. Source code can be found at https://github.com/Delay-Xili/LDR (accessed on 9 February 2022).

### 1.1. Learning Generative Models via Auto-Encoding or GAN

**Auto-Encoding and its variants.** In the machine-learning literature, roughly speaking, there have been two representative approaches to such a distribution-learning task. One is the classic “Auto Encoding” (AE) approach [3,4] that aims to simultaneously learn an encoding mapping *f* from x to z and an (inverse) decoding mapping *g* from z back to x:(2)$$X\overset{\;f(x,\theta )\;}{\to }Z\overset{\;g(z,\eta )\;}{\to }\hat{X}.$$

Here, we use bold capital letters to indicate a matrix of finite samples $X=\left[x^{1},\ldots ,x^{n}\right]\in R^{D\times n}$ of x and their mapped features $Z=\left[z^{1},\ldots ,z^{n}\right]\subset R^{d\times n}$, respectively. Typically, one wishes for two properties: firstly, the decoded samples $\hat{X}$ are “similar” or close to the original X, say in terms of maximum likelihood p(X); and secondly, the (empirical) distribution of the mapped samples Z, denoted as $\hat{p}\left(z|X\right)$, is close to certain desired prior distribution p(z), say some much lower-dimensional multivariate Gaussian (The classical PCA can be viewed as a special case of this task. In fact, the original auto-encoding is precisely cast as *nonlinear* PCA [3], assuming the data lie on only one nonlinear submanifold M).

However it is typically very difficult, often computationally intractable to maximize the likelihood function p(X) or to minimize certain “distance”, say the *KL-divergence*$D_{KL}\left(\hat{p},p\right)$, between $\hat{p}\left(z|X\right)$ and p(z). Except for simple distributions such as Gaussian, the KL divergence usually does not have a closed-form, even for a mixture of Gaussians. The likelihood and the KL-divergence become ill-conditioned when the supports of the distributions are low-dimensional (i.e., degenerate) and not overlapping (which is almost always the case in practice when dealing with distributions of high-dimensional data in high-dimensional spaces). So in practice, one typically chooses to minimize instead certain approximate bounds or surrogates derived with various simplifying assumptions on the distributions involved, as is the case in variational auto-encoding (VAE) [5,6]. As a result, even after learning, the precise posterior distribution of $\hat{p}\left(z|X\right)$ remains unclear or hidden inside the feature space.

In this work, we will show that if we impose specific requirements on the (distribution of) learned feature z to be a mixture of subspace-like Gaussians, a natural closed-form distance can be introduced for such distributions based on rate distortion from the information theory. In addition, the optimal solution to the feature representation within this family can be learned directly from the data *without specifying any target p(z) in advance*, which is particularly difficult in practice when the distribution of a mixed dataset is multi-modal and each component may have a different dimension.

**GAN and its variants.** Compared to measuring distribution distance in the (often controlled) feature space z, a much more challenging issue with the above auto-encoding approach is how to effectively measure the distance between the decoded samples $\hat{X}$ and the original X in the data space x. For instance, for visual data such as images, their distributions p(X) or generative models p(X|z) are often not known. Despite extensive studies in the computer vision and image processing literature [7], it remains elusive to find a good measure for similarity of real images that is both efficient to compute and effective in capturing visual quality and semantic information of the images equally well. Precisely due to such difficulties, it has been suggested early on by [8] that one may have to take a discriminative approach to learn the distribution or a generative model for visual data. More recently, *Generative Adversarial Nets (GAN)* [9] offers an ingenious idea to alleviate this difficulty by utilizing a powerful discriminator *d*, usually modeled and learned by a deep network, to discern differences between the generated samples $\hat{X}$ and the real ones X:(3)$$Z\overset{\;g(z,\eta )\;}{\to }\hat{X},\;X\overset{\;d(x,\theta )\;}{\to }0,1.$$

To a large extent, such a discriminator plays the role of minimizing certain distributional distance, e.g., the *Jensen–Shannon divergence*, between the data X and $\hat{X}$. Compared to the KL-divergence, the JS-divergence is well-defined even if the supports of the two distributions are non-overlapping. (However, JS-divergence does not have a closed-form expression even between two Gaussians, whereas KL-divergence does). However, as shown in [10], since the data distributions are low-dimensional, the JS-divergence can be highly ill-conditioned to optimize. (This may explain why many additional heuristics are typically used in many subsequent variants of GAN). So, instead, one may choose to replace JS-divergence with the earth mover’s distance or the Wasserstein distance. However both JS-divergence and W-distance can only be approximately computed between two general distributions. (For instance, the W-distance requires one to compute the maximal difference between expectations of the two distributions over all 1-Lipschitz functions). Furthermore, neither the JS-divergence nor the W-distance have closed-form formulae, even for the Gaussian distributions. (The ($\ell^{1}$-norm) W-distance can be bounded by the ($\ell^{2}$-norm) W2-distance which has a closed-form [11]. However, as is well-known in high-dimensional geometry, $\ell^{1}$-norm and $\ell^{2}$ norm deviate significantly in terms of their geometric and statistical properties as the dimension becomes high [12]. The bound can become very loose). However, from a data representation perspective, *subspace-like Gaussians (e.g., PCA) or a mixture of them are the most desirable family of distributions that we wish our features to become.* This would make all subsequent tasks (generative or discriminative) much easier. In this work, we will show how to achieve this with a different fundamental metric, known as the rate reduction, introduced by [13].

The original GAN aims to directly learn a mapping g(·), called a generator, from a standard distribution (say, a low-dimensional Gaussian random field) to the real (visual) data distribution in a high-dimensional space. However, distributions of real-world data can be rather sophisticated and often contain *multiple* classes and *multiple* factors in each class [14]. This makes learning the mapping *g* rather challenging in practice, suffering difficulties such as *mode-collapse* [15]. As a result, many variants of GAN have been subsequently developed in order to improve the stability and performance in learning multiple modes and disentangling different factors in the data distribution, such as *Conditional GAN* [16,17,18,19,20], *InfoGAN* [21,22], or *Implicit Maximum Likelihood Estimation (IMLE)* [23,24]. In particular, to learn a generator for multi-class data, prevalent conditional GAN literature requires label information as conditional inputs [16,25,26,27]. Recently, [28,29] has proposed training a *k*-class GAN by generalizing the two-class cross entropy to a (k+1)-class cross entropy. In this work, *we will introduce a more refined 2k-class measure* for the *k* real and *k* generated classes. In addition, to avoid features for each class collapsing to a singleton [30], instead of cross entropy, *we will use the so-called rate-reduction measure that promotes multi-mode and multi-dimension in the learned features* [13]. One may view the rate reduction as a metric distance that has closed-form formulae for a mixture of (subspace-like) Gaussians, whereas neither JS-divergence nor W-distance can be computed in closed form (even between two Gaussians).

Another line of research is about how to stabilize the training of GAN. SN-GAN [31] has shown that spectral normalization on the discriminator is rather effective, which we will adopt in our work, although our formulation is not so sensitive to such choice designed for GAN (see ablation study in [Sec app1dot9-entropy-24-00456]). PacGAN [32] shows that the training stability can be significantly improved by packing a pair of real and generated images together for the discriminator. Inspired by this work, *we show how to generalize such an idea to discriminating an arbitrary number of pairs of real and decoded samples without concatenating the samples.* Our results in this work will even suggest that the larger the batch size discriminated, the merrier (see ablation study in [Sec app1dot10-entropy-24-00456]). In addition, ref. [29] has shown that optimizing the latent features leads to state-of-the-art visual quality. Their method is based on the deep compressed sensing GAN [28]. Hence, there are strong reasons to believe that their method essentially utilizes the *compressed sensing* principle [12] to implicitly exploit the low-dimensionality of the feature distribution. Our framework *will explicitly expose and exploit such low-dimensional structures on the learned feature distribution.*

**Combination of AE and GAN.** Although AE (VAE) and GAN originated with somewhat different motivations, they have evolved into popular and effective frameworks for learning and modeling complex distributions of many real-world data such as images. (In fact, in some idealistic settings, it can be shown that AE and GAN are actually equivalent: for instance, in the LOG settings, authors in [33] have shown that GAN coincides with the classic PCA, which is precisely the solution to auto-encoding in the linear case). Many recent efforts tend to combine both auto-encoding and GAN to generate more powerful generative frameworks for more diverse data sets, such as [15,34,35,36,37,38,39,40,41,42]. As we will see, in our framework, AE and GAN can be naturally interpreted as two different segments of a closed-loop data transcription process. However, unlike GAN or AE (VAE), the “origin” or “target” distribution of the feature z will no longer be specified *a priori*, and is instead learned from the data x. In addition, *this intrinsically low-dimensional distribution of z (with all of its low-dimensional supports) is explicitly modeled as a mixture of orthogonal subspaces (or independent Gaussians) within the feature space $R^{d}$*, sometimes known as the principal subspaces.

**Universality of Representations.** Note that GANs (and most VAEs) are typically designed without explicit modeling assumptions on the distribution of the data nor on the features. Many even believe that it is this “universal” distribution learning capability (assuming minimizing distances between arbitrary distributions in high-dimensional space can be solved efficiently, which unfortunately has many caveats and often is impractical) that is attributed to their empirical success in learning distributions of complicated data such as images. In this work, we will provide empirical evidence that such an “arbitrary distribution learning machine” might not be necessary. (In fact, it may be computationally intractable in general). A *controlled and deformed* family of low-dimensional linear subspaces (Gaussians) can be more than powerful, and expressive enough to model real-world visual data. (In fact, a Gaussian mixture model is already a universal approximator of almost arbitrary densities [43]. Hence, we do not loose any generality at all). As we will also see, once we can place a proper and precise metric on such models, the associated learning problems can become much better conditioned and more amenable to rigorous analysis and performance guarantees in the future.

### 1.2. Learning Linear Discriminative Representation via Rate Reduction

Recently, the authors in [2] proposed a new objective for deep learning that aims to learn a *linear discriminative representation* (LDR) for multi-class data. The basic idea is to map distributions of real data, potentially on *multiple* nonlinear submanifolds $\cup_{j=1}^{k}M_{j}\subset R^{D}$ (in classical statistical settings, such nonlinear structures of the data were also referred to as principal curves or surfaces [44,45]. There has been a long quest of trying to extend PCA to handle potential nonlinear low-dimensional structures in data distribution (see [46] for a thorough survey) to a family of canonical models consisting of multiple independent (or orthogonal) linear subspaces, denoted as $\cup_{j=1}^{k}S_{j}\subset R^{d}$. To some extent, this generalizes the classic nonlinear PCA [3] to more general/realistic settings where we simultaneously apply *multiple nonlinear PCAs* to data on multiple nonlinear submanifolds. Or equivalently, the problem can also be viewed as a nonlinear extension to the classic *Generalized PCA* (GPCA) [46]. (Conventionally, “generalized PCA” refers to generalizing the setting of PCA to multiple *linear* subspaces. Here, we need to further generalize multiple *nonlinear* submanifolds. Unlike conventional discriminative methods that only aim to predict class labels as one-hot vectors, the LDR aims to learn the likely multi-dimensional distribution of the data, hence it is suitable for both discriminative and generative purposes. It has been shown that this can be achieved via maximizing the so-called “rate reduction” objective based on the rate distortion of subspace-like Gaussians [47].

**LDR via MCR${}^{2}$.** More precisely, consider a set of data samples $X=\left[x^{1},\ldots ,x^{n}\right]\in R^{D\times n}$ from *k* different classes. That is, we have $X=\cup_{j=1}^{k}X_{j}$ with each subset of samples $X_{j}$ belonging to one of the low-dimensional submanifolds: $X_{j}\subset M_{j},j=1,\ldots ,k$. Following the notation in [2], we use a matrix $\prod^{j}\left(i,i\right)=1$ to denote the membership of sample *i* belonging to class *j* (and $\prod^{j}=0$ otherwise). One seeks a continuous mapping f(·,θ):x↦z from X to an optimal representation $Z=\left[z^{1},\ldots ,z^{n}\right]\subset R^{d\times n}$:(4)$$X\overset{\;f(x,\theta )\;}{\to }Z,$$
which maximizes the following coding rate-reduction objective, known as *the MCR${}^{2}$ principle* [13]:(5)$$\begin{array}{cc} \max_{Z}\;\Delta R\left(Z\;|\;\prod ,\epsilon \right)\; & ≐\underset{R(Z\;|\epsilon )}{\underbrace{\frac{1}{2}\log\det\left(I+\alpha ZZ^{*}\right)}}\;-\;\underset{R_{c}\left(Z\;|\prod ,\epsilon \right)}{\underbrace{\sum_{j=1}^{k}\frac{\gamma_{j}}{2}\log\det\left(I+\alpha_{j}Z\prod^{j}Z^{*}\right)}}, \end{array}$$
where $\alpha =\frac{d}{n\epsilon^{2}}$, $\alpha_{j}=\frac{d}{\mathrm{tr}\left(\prod^{j}\right)\epsilon^{2}}$, $\gamma_{j}=\frac{\mathrm{tr}(\prod^{j})}{n}$ for j=1,…,k. In this paper, for simplicity we denote ΔR(Z|∏,ϵ) as ΔR(Z) assuming ∏,ϵ are known and fixed. The first term R(Z|ϵ), or R(Z) for short, is the coding rate of the whole feature set Z (coded as a Gaussian source) with a prescribed precision ϵ; the second term $R_{c}\left(Z\;|\prod ,\epsilon \right)$, or simply $R_{c}\left(Z\right)$, is the average coding rate of the *k* subsets of features $Z_{j}=f\left(X_{j}\right)$ (each coded as a Gaussian).

As has been shown by [13], maximizing the difference between the two terms will expand the whole feature set while compressing and linearizing features of each of the *k* classes. If the mapping *f* maximizes the rate reduction, it maps the features of different classes into independent (orthogonal) subspaces in $R^{d}$. Figure 1 illustrates a simple example of data with k=2 classes (on two submanifolds) mapped to two incoherent subspaces (solid black lines). Notice that, compared to AE (Equation 2) and GAN (Equation 3), the above mapping (Equation 4) is only one-sided: from the data X to the feature Z. In this work, we will see how to use the rate-reduction metric to establish inverse mapping from the feature Z back to the data X, while still preserving the subspace structures in the feature space.

## 2. Data Transcription via Rate Reduction

### 2.1. Closed-Loop Transcription to an LDR (CTRL)

One issue with this one-sided LDR learning (Equation 4) is that maximizing the above objective (Equation 5) tends to expand the dimension of the learned subspace for features in each class (if the dimension of the feature space *d* is too high, maximizing the rate reduction may over-estimate the dimension of each class. Hence, to learn a good representation, one needs to pre-select a proper dimension for the feature space, as achieved in the experiments in [13]. In fact the same “model selection” problem persists even in the simplest single-subspace case, which is the classic PCA [48]. Selecting the correct number of principal components in a heterogeneous noisy situation remains an active research topic [49]). To verify whether the learned features are neither over-estimating nor under-estimating the data structure, we may consider learning a decoder g(·,η):z↦x from the representation Z=f(X,θ) back to the data space x: $\hat{X}=g\left(Z,\eta \right)$, and check how close X and $\hat{X}$ are or how close their features Z and $\hat{Z}=f\left(\hat{X},\theta \right)$ are. In principle, the decoder *g* should examine if all the learned features by the encoder *f* are both necessary and sufficient for achieving this task. The overall pipeline can be illustrated by the following “closed-loop” diagram:(6)$$X\overset{\;f(x,\theta )\;}{\to }Z\overset{\;g(z,\eta )\;}{\to }\hat{X}\overset{\;f(x,\theta )\;}{\to }\;\hat{Z},$$
where the overall model has parameters: Θ={θ,η}.

Notice that in the above process, the segment from X to $\hat{X}$ resembles a typical *Auto-Encoding* process; although, as we will soon see, our MCR${}^{2}$-based encoder *f* plays an additional role as a discriminator. The segment from Z to $\hat{Z}$ draws resemblance to the typical GAN process; although, in our context, the distribution of the latent variable z will be learned from the data x. Despite these connections, as we will soon see, this new closed-loop formulation will allow us to utilize the *error feedback* mechanism (widely practiced in control systems) and directly enforce loop consistency between encoding and decoding (networks) *without* using any additional discriminator(s) that are typically needed in existing VAE/GAN architectures.

Here, in the specific context of rate reduction, we name this special auto-encoding process “*Transcription to an LDR*” since the maximal rate-reduction principle explicitly transcribes the data X, via *f*, to features Z on a linear discriminative representation (LDR) (through our extensive experiments on diverse real-world visual datasets, one does not lose any generality or expressiveness by restricting to this special but rich class of models. On the contrary, the restriction significantly simplifies and improves the learning process), which can be subsequently decoded back to the data space $\hat{X}$, via *g*. Hence, the encoding and decoding maps *f* and *g* together form a “closed-loop” process, as illustrated in Figure 1. We hope that this closed-loop transcription to an LDR (CTRL) has the following good properties:**Injectivity:** the generated $\hat{x}=g\left(f\left(x,\theta \right),\eta \right)\in \hat{X}$ should be as close to (ideally the same as) the original data x∈X, in terms of certain measures of similarity or distance.**Surjectivity:** for all mapped images z=f(x)∈Z of the training data x∈X, there are decoded samples $\hat{z}=f\left(g\left(z,\eta \right),\theta \right)\in \hat{Z}$ close to (ideally the same as) z.

Mathematically, we seek an *embedding* of the data x supported on certain nonlinear submanifolds $\cup_{j=1}^{k}M_{j}$ in the space $R^{D}$ to feature z on a set of (discriminative) linear subspaces $\cup_{j=1}^{k}S_{j}$ in the feature space $R^{d}$. Ideally, both *f* and *g* should be embeddings [1], when restricted on the support of the data distribution or that of the features. (That is, we hope $f{|}_{M_{j}}$ and $g{|}_{S_{j}}$ are all embeddings for all j=1,…,k.) In addition, more ideally, we hope *f* and *g* are mutually inverse embeddings: g∘f=Id (when restricted on the submanifolds). Nevertheless, if we are only interested in learning the distribution, embeddings of the support would often suffice the purposes (e.g., classification or generative purposes). Notice that the above goals are similar to many VAE+GAN-related methods in the machine-learning literature, such as BiGAN [38] and ALI [39]. We will discuss the differences of our approach from these existing methods in Section 2.3 (as well as providing some experimental comparisons in the Appendix A).

At first sight, this is a rather daunting task, since we are trying to learn over a (seemingly infinite-dimensional) functional space of all embeddings and distributions from finite samples. In this work, we will take a more pragmatic approach and show how one can learn a good encoding, decoding, and representation tuple: (f,g,z) from X via tractable computational means. In particular, we will convert the above goals to certain feasible programs that optimize a sensible measure of goodness for the learned representations Z.

### 2.2. Measuring Distances in the Feature Space and Data Space

**Contractive measure for the decoder.** For the *second* item in the above wishlist, as the representations in the feature space z are by design linear subspaces or (degenerate) Gaussians, we have geometrically or statistically meaningful metrics for both samples and distributions in the feature space z. For example, we care about the distance between distributions between the features of the original data Z and the transcribed $\hat{Z}$. Since the features of each class, $Z_{j}$ and ${\hat{Z}}_{j}$, are similar to subspaces/Gaussians, their “distance” can be measured by the rate reduction, with (Equation 5) restricted to two sets of equal size:(7)$$\Delta R(Z_{j},{\hat{Z}}_{j})≐R(Z_{j}\cup {\hat{Z}}_{j})-\frac{1}{2}(R(Z_{j})+R({\hat{Z}}_{j})).$$

According to the interpretation of the rate reduction given in [13], the above quantity precisely measures the volume of the space between $Z_{j}$ and ${\hat{Z}}_{j}$, illustrated as a pair of black and blue lines in Figure 1. Then, for the “distance” of all, say *k*, classes, we simply sum the rate reduction for all pairs:(8)$$d\left(Z,\hat{Z}\right)≐\min_{\eta }\sum_{j=1}^{k}\Delta R(Z_{j},{\hat{Z}}_{j})=\min_{\eta }\sum_{j=1}^{k}\Delta R(Z_{j},f\left(g\left(Z_{j},\eta \right),\theta \right)),$$
where $Z_{j}=f\left(X_{j},\theta \right)$ and ${\hat{Z}}_{j}=f\left({\hat{X}}_{j},\theta \right)$. Obviously, a main goal of the learned decoder g(·,η) is to *minimize* the distance between these distributions. Notice that if the encoder *f* preserves (i.e., injective for) the intrinsic structures of the original data X, (this is typically the case for MCR${}^{2}$-based feature representation [13]) this criterion essentially aims to ensure there will be some decoded sample $\hat{x}$ close to every data sample x—hence the decoder *g* should be “surjective”. According to the ideas of IMLE [23], such a requirement could effectively help to avoid mode-collapsing or mode-dropping.

**Contrastive measure for the encoder.** For the *first* item in our wishlist, however, we normally do not have a natural metric or “distance” for similarity of samples or distributions in the original data space x for data such as images. As mentioned before, finding proper metrics or distance functions on natural images has always been an elusive and challenging task [7]. To alleviate this difficulty, we can measure the similarity or difference between $\hat{X}$ and X through their mapped features $\hat{Z}$ and Z in the feature space (again assuming *f* is structure-preserving). If we are interested in discerning *any* differences in the distributions of the original and transcribed samples, we may view the MCR${}^{2}$ feature encoder f(·,θ) as a “discriminator” to *magnify* any difference between all pairs of $X_{j}$ and ${\hat{X}}_{j}$, by simply maximizing, instead of minimizing, the *same quantity* in (Equation 8):(9)$$d\left(X,\hat{X}\right)≐\max_{\theta }\sum_{j=1}^{k}\Delta R(Z_{j},{\hat{Z}}_{j})=\max_{\theta }\sum_{j=1}^{k}\Delta R(f\left(X_{j},\theta \right),f\left({\hat{X}}_{j},\theta \right)).$$

That is, a “distance” between X and $\hat{X}$ can be measured as the maximally achievable rate reduction between all pairs of classes in these two sets. In a way, this measures how well or badly the decoded $\hat{X}$ aligns with the original data X—hence measuring the goodness of “injectivity” of the encoder *f*. Notice that such a discriminative measure is consistent with the idea of GAN [9] that tries to separate X and $\hat{X}$ into two classes, measured by the cross-entropy. Nevertheless, here the MCR${}^{2}$-based discriminator *f* naturally generalizes to cases when the data distributions are multi-class and multi-modal, and the discriminativeness is measured with a more refined measure—the rate reduction—instead of the typical two-class loss (e.g., cross entropy) used in GANs. See [Sec app1dot8-entropy-24-00456] for comparisons with some ablation studies.

One may wonder why we need the mapping f(·,θ) to function as a discriminator between X and $\hat{X}$ by maximizing $\max_{\theta }\Delta R(f\left(X,\theta \right),f\left(\hat{X},\theta \right))$. Figure 2 gives a simple illustration: there might be many decoders *g* such that f∘g is an identity (Id) mapping. Here, we use the notion of “identity mapping” in a loose sense: depending on the context, it could simply mean an embedding from $S_{z}$ to $S_{z}$. f∘g(z)=z for all z in the subspace $S_{z}$ in the feature space. However, g∘f is not necessarily an auto-encoding map for x in the original distribution $S_{x}$ (here for simplicity drawn as a subspace). That is, $g\circ f\left(S_{x}\right)⊄S_{x}$, let alone $g\circ f\left(S_{x}\right)=S_{x}$ or g∘f(x)=x. One should expect, without careful control of the image of *g*, with high probability, this would be the case, especially when the support of the distribution of x is extremely low-dimensional in the original high-dimensional data space. For example, as we will see in the experiments, the intrinsic dimension of the submanifold associated with each image category is about a dozen, whereas images are embedded in a (pixel) space of thousands or tens of thousands of dimensions.

**Remark: representing the encoding and decoding mappings.** Some practical questions arise immediately: how rich should the families of functions be that we should consider to use for the encoder *f* and decoder *g* that can optimize the above rate-reduction-type objectives? In fact, similar questions exist for the formulation of GAN, regarding the realizability of the data distribution by the generator, see [50]. Conceptually, here we know that the encoder *f* needs to be rich enough to discriminate (small) deviations from the true data support $M_{j}$, while the decoder *g* needs to be expressive enough to generate the data distribution from the learned mixture of subspace-Gaussians. How should we represent or parameterize them, hence making our objectives computable and optimizable? For the most general cases, these remain widely open and challenging mathematical and computational problems. As we mentioned earlier, in this work, we will take a more pragmatic approach by simply representing these mappings with popular neural networks that have empirically proven to be good at approximating distributions of practical (visual) datasets or for achieving the maximum of the rate-reduction-type objectives [13]. Nevertheless, our experiments indicate that our formulation and objectives are *not so sensitive* to particular choices in network structures or many of the tricks used to train them. In addition, in the special cases when the real data distribution is benignly deformed from an LDR, the work of [2] has shown that one can explicitly construct these mappings from the rate-reduction objectives in the form of a deep network known as ReduNet. However, it remains unclear how such constructions could be generalized to closed-loop settings. Regardless, answers to these questions are beyond the scope of this work, as our purposes here are mainly to empirically verify the validity of the proposed closed-loop data transcription framework.

### 2.3. Encoding and Decoding as a Two-Player MiniMax Game

Comparing the contractive and contrastive nature of (Equation 8) and (Equation 9) on the same utility, we see the roles of the encoder f(·,θ) and the decoder g(·,η) naturally as “**a two-player game**”: *while the encoder f tries to magnify the difference between the original data and their transcribed data, the decoder g aims to minimize the difference.* Now for convenience, let us define the “closed-loop encoding” function:(10)$$h\left(x,\theta ,\eta \right)≐f\left(g\left(f(x,\theta ),\eta \right),\theta \right):\;x\mapsto z.$$

Ideally, we want this function to be very close to f(x,θ) or at least the distributions of their images should be close. With this notation, combining (Equation 8) and (Equation 9), a closed-loop notion of “distance” between X and $\hat{X}$ can be computed as *an equilibrium point* to the following Min-Max (or Max-Min) program for the same utility in terms of rate reduction (theoretically, there might be significant difference in formulating and seeking the desired solution as the equilibrium point to a min-max or max-min game. In practice, we do not see major differences as we optimize the program by simply alternating between minimization and maximization. We leave a more careful investigation to future work):(11)$$D\left(X,\hat{X}\right)≐\min_{\eta }\max_{\theta }\sum_{j=1}^{k}\Delta R\left(f\left(X_{j},\theta \right),h\left(X_{j},\theta ,\eta \right)\right).$$

Notice that this only measures the difference between (features of) the original data and its transcribed version. It does not measure how good the representation Z (or $\hat{Z}$) is for the multiple classes within X (or $\hat{X}$). To this end, we may combine the above distance with the original MCR${}^{2}$-type objectives (Equation 5): namely, the rate reduction ΔR(Z) and $\Delta R(\hat{Z})$ for the learned LDR Z for X and $\hat{Z}$ for the decoded $\hat{X}$. Notice that although the encoder *f* tries to *maximize* the multi-class rate reduction of the features Z of the data X, the decoder *g* should *minimize* the rate reduction of the multi-class features $\hat{Z}$ of the decoded $\hat{X}$. That is, the decoder *g* tries to use a minimal coding rate needed to achieve a good decoding quality.

Hence, the overall “multi-class” Min-Max program for learning the Closed-loop Transcription to an LDR, named CTRL-Multi, is subject to certain constraints (upper or lower bounds) on the first term and the second term. In this work, we only consider the simple case by adding these rate-reduction quantities together. Of course, in the future, one may consider other more delicate formulations. For instance, we may consider a Min-Max game on the third term (Equation 11). Such constrained minimax games have also started to draw attention lately [51].
(12)$$\begin{array}{ccc} \min_{\eta }\max_{\theta }T_{X}\left(\theta ,\eta \right) & ≐ & \underset{\mathrm{Expansive}\;\mathrm{encode}}{\underbrace{\Delta R\left(f(X,\theta )\right)}}+\underset{\mathrm{Compressive}\;\mathrm{decode}}{\underbrace{\Delta R\left(h(X,\theta ,\eta )\right)}}+\sum_{j=1}^{k}\underset{\mathrm{Contrastive}\;\mathrm{encode}\;\&\;\mathrm{Contractive}\;\mathrm{decode}}{\underbrace{\Delta R\left(f\left(X_{j},\theta \right),h\left(X_{j},\theta ,\eta \right)\right)}}\\ & = & \Delta R\left(Z(\theta )\right)+\Delta R\left(\hat{Z}\left(\theta ,\eta \right)\right)+\sum_{j=1}^{k}\Delta R\left(Z_{j}\left(\theta \right),{\hat{Z}}_{j}\left(\theta ,\eta \right)\right). \end{array}$$

Empirically, we have evaluated the necessity of these terms in an ablation study (see [Sec app1dot8dot3-entropy-24-00456]). Notice that, without the terms associated with the generative part *h* or with all such terms fixed as constant, the above objective is precisely the original MCR${}^{2}$ objective proposed by [13]. In an unsupervised setting, if we view each sample (and its augmentations) as its own class, the above formulation remains exactly the same. The number of classes *k* is simply the number of independent samples. In addition, notice that the minimax objective function depends only on (features of) the data X, hence one can learn the encoder and decoder (parameters) without the need for sampling or matching any additional distribution (as typically needed in GANs or VAEs).

As a special case, if X only has one class, the above Min-Max program reduces (as the first two rate reduction terms automatically become zero) to a special “two-class” or “binary” form, named CTRL-Binary, between X and the decoded $\hat{X}$ by viewing X and $\hat{X}$ as two classes {0,1}. Notice that this binary case resembles formulation of the original GAN (Equation 3). Nevertheless, instead of using cross entropy, our formulation adopts a more refined rate-reduction measure, which has been shown to promote diversity in the learned representation [13]).
(13)$$\mathrm{CTRL}\text{-}\mathrm{Binary}:\;\min_{\eta }\max_{\theta }T_{X}^{b}\left(\theta ,\eta \right)≐\Delta R\left(f(X,\theta ),h(X,\theta ,\eta )\right)=\Delta R(Z\left(\theta \right),\hat{Z}\left(\theta ,\eta \right)).$$

Sometimes, even when X contains multiple classes/modes, one could still view all classes together as one class. Then, the above binary objective is to align the union distribution of all classes with their decoded $\hat{X}$. This is typically a simpler task to achieve than the multi-class one (Equation 12), since it does not require learning of a more refined multi-class CTRL for the data, as we will later see in experiments. Notice that one good characteristic of the above formulation is that *all quantities in the objectives are measured in terms of rate reduction for the learned features* (assuming features eventually become subspace Gaussians).

In all of our subsequent experiments, we solve the above minimax programs using the most basic gradient descent–ascent (GDA) algorithm [52] that alternates between the minimization and maximization, with the same learning rate and without any timescale separation (as typically needed for training GANs [53]). Although more refined optimization schemes can likely further improve the efficiency and performance, we leave these for future investigations.

**Remark: closed-loop error correction.** One may notice that our framework (see Figure 1) draws inspiration from closed-loop error correction widely practiced in feedback control systems. In the machine-learning and deep-learning literature, the idea of closed-loop error correction and closed-loop fixed point has been explored before to interpret the recursive error-correcting mechanism and explain stability in a forward (predictive) deep neural network, for example the *deep equilibrium networks* [54] and the *deep implicit networks* [55], again drawing inspiration from feedback control. Here, in our framework, the closed-loop mechanism is not used to interpret the encoding or decoding (forward) networks *f* and *g*. Instead, it is used to form an overall feedback system between the two encoding and decoding networks for correcting the “error” in the distributions between the data x and the decoded $\hat{x}$. Using terminology from control theory, one may view the encoding network *f* as a “sensor” for error feedback while the decoding network *g* as a “controller” for error correction. However, notice that here the “target” for control is not a scalar nor a finite dimensional vector, but a continuous mapping—in order for the distribution of $\hat{x}$ to match that of the data x. This is in general a control problem in an infinite dimensional space. The space of diffeomorphisms of submanifolds is infinite-dimensional [1]. Ideally, we hope when the sensor *f* and the controller *g* are optimal, the distribution of x becomes a “fixed point” for the closed loop while the distribution of z reaches a compact LDR. Hence, the minimax programs (Equation 12) and (Equation 13) can also be interpreted as games between an error-feedback sensor and an error-reducing controller.

**Remark: relation to bi-directional or cycle consistency.** The notion of “bi-directional” and “cycle” consistency between encoding and decoding has been exploited in the works of BiGAN [38] and ALI [39] for mappings between the data and features and in the work of CycleGAN [56] for mappings between two different data distributions. In our context, it is similar in order to promote g∘f and f∘g to be close to identity mappings (either for the distributions or for the samples). Interestingly, our new closed-loop formulation actually “decouples” the data X, say, observed from the external world, from their internally represented features Z. The objectives (Equation 12) and (Equation 13) are functions of *only* the internal features Z(θ) and $\hat{Z}\left(\theta ,\eta \right)$, which can be learned and optimized by adjusting the neural networks f(·,θ) and g(·,η) alone. There is no need for any additional external metrics or heuristics to promote how “close” the decoded images $\hat{X}$ are to X. This is very different from most VAE/GAN-type methods such as BiGAN and ALI that require additional discriminators (networks) for the images and the features. Some experimental comparison are given in the [Sec app1dot2-entropy-24-00456]. In addition, in [Sec app1dot8dot1-entropy-24-00456], we provide some ablation study to illustrate the importance and benefit of a closed loop for enforcing the consistency between the encoder and decoder.

**Remark: transparent versus hidden distribution of the learned features.** Notice that in our framework, there is no need to explicitly specify a prior distribution either as a target distribution to map to for AE (Equation 2) or as an initial distribution to sample from for GAN (Equation 3). The common practice in AEs or GANs is to specify the prior distribution as a generic Gaussian. This is however particularly problematic when the data distribution is multi-modal and has multiple low-dimensional structures, which is commonplace for multi-class data. In this case, the common practice in AEs or GANs is to train a conditional GAN for different classes or different attributes. However, here we only need to assume the desired target distribution belonging to the family of LDRs. The specific optimal distribution of the features within this family is then learned from the data directly, and then can be represented *explicitly* as a mixture of independent subspace Gaussians (or equivalently, a mixture of PCAs on independent subspaces). We will give more details in the experimental Section 3 as well as more examples in [Sec app1dot2-entropy-24-00456], [Sec app1dot3-entropy-24-00456] and [Sec app1dot4-entropy-24-00456]. Although many GAN + VAE-type methods can learn bidirectional encoding and decoding mappings, the distribution of the learned features inside the feature space remains *hidden* or even *entangled*. This makes it difficult to sample the feature space for generative purposes or to use the features for discriminative tasks. (For instance, typically one can only use so-learned features for nearest-neighbor-type classifiers [38], instead of nearest subspace as in this work, see Section 3.3).

## 3. Empirical Verification on Real-World Imagery Datasets

This experiment section serves three purposes: First, we empirically justify the proposed formulation for data transcription by demonstrating good properties of the learned encoder, decoder, and representation tuple (f,g,z) from X. Second, we compare our method with several representative methods from the GAN family and VAE family. The purpose of the comparison is *not* to compete for any state-of-the-art performance. Instead, we want to convincingly verify the validity of the proposed framework and its potential in going beyond. Finally, we evaluate the so-learned CTRL through both generative tasks (controlled visualization) and discriminative (classification) tasks. More extensive experimental results, evaluations, and ablation studies can be found in the Appendix A.

**Datasets.** We provide extensive qualitative and quantitative experimental results on the following datasets: MNIST [57], CIFAR-10 [58], STL-10 [59], CelebA [60], LSUN bedroom [61], and ImageNet ILSVRC 2012 [62]. The network architectures and implementation details can be found in [Sec app1dot1-entropy-24-00456] and corresponding Appendix A for each dataset.

### 3.1. Empirical Justification of CTRL Transcription

To empirically validate our new framework, we conduct experiments from a small low-variety dataset (MNIST), to a small dataset of diverse real-world objects (CIFAR-10), to higher resolution images (STL-10, CelebA, LSUN-bedroom), to a large-scale diverse image set (ImageNet). The results are evaluated both quantitatively and qualitatively. Implementation details, more experimental results, and ablation studies are given in Appendix A.

**Comparison (IS and FID) with other formulations.** First, we conduct five experiments to fairly compare our formulation with GAN [63] and VAE(-GAN) [64] on MNIST and CIFAR-10. Except for the objective function, everything else is exactly the same for all methods (e.g., networks, training data, optimization method). These experiments are: (1). GAN; (2). GAN with its objective replaced by that of the CTRL-Binary (Equation 13); (3). VAE-GAN; (4). Binary CTRL (Equation 13); and (5). Multi-class CTRL (Equation 12). Some visual comparison is given in Figure 3. IS [65] and FID [66] scores are summarized in Table 1. Here, for simplicity, we have chosen a uniform feature dimension d=128 for all datasets. If we choose a higher feature dimension, say d=512, for the more complex CIFAR-10 dataset, the visual quality can be further improved, see Table A14 in [Sec app1dot11-entropy-24-00456].

As we see from Table 1, replacing cross-entropy with the Equation (Equation 13) can improve the generative quality. The two CTRL formulations are clearly on par with the others in terms of IS and significantly better in FID. Finally, with the same training datasets, the quality of CTRL-Multi is lower than that of CTRL-Binary. This is expected, as the multi-class task is more challenging. Nevertheless, as we will see soon, images decoded by CTRL-Multi align much better with their classes than Binary.

Visualizing correlation of features Z and decoded features $\hat{Z}$. We visualize the cosine similarity between Z and $\hat{Z}$ learned from the multi-class objective (Equation 12) on MNIST, CIFAR-10 and ImageNet (10 classes), which indicates how close $\hat{z}=f\circ g\left(z\right)$ is from z. Results in Figure 4 show that Z and $\hat{Z}$ are aligned very well within each class. The block-diagonal patterns for MNIST are sharper than those for CIFAR-10 and ImageNet, as images in CIFAR-10 and ImageNet have more diverse visual appearances.

**Visualizing auto-encoding of the data X and the decoded $\hat{X}$.** We compare some representative X and $\hat{X}$ on MNIST, CIFAR-10 and ImageNet (10 classes) to verify how close $\hat{x}=g\circ f\left(x\right)$ is to x. The results are shown in Figure 5, and visualizations are created from training samples. Visually, the auto-encoded $\hat{x}$ faithfully captures major visual features from its respective training sample x, especially the pose, shape, and layout. For the simpler dataset such as MNIST, auto-encoded images are almost identical to the original. The visual quality is clearly better than other GAN+VAE-type methods, such as VAE-GAN [34] and BiGAN [38]. We refer the reader to [Sec app1dot2-entropy-24-00456], [Sec app1dot4-entropy-24-00456] and [Sec app1dot7-entropy-24-00456] for more visualization of results on these datasets, including similar results on transformed MNIST digits. More visualization results for learned models on real-life image datasets such as STL-10, CeleB, and LSUN can be found in the [Sec app1dot5-entropy-24-00456] and [Sec app1dot6-entropy-24-00456].

### 3.2. Comparison to Existing Generative Methods

Table 2 gives a quantitative comparison of visual quality of our method with others on CIFAR-10, STL-10, and ImageNet. In general, there is a large difference in terms of FID and IS scores between the GAN family and the VAE family of models. SNGAN [31] are commonly used methods in most generative applications, while LOGAN [29] is the state-of-the-art method on ImageNet in terms of FID and IS. More comparisons with existing methods, including results on on the higher-resolution ImageNet dataset, can be found in Table A10 of the [Sec app1dot7-entropy-24-00456].

As we see, even if the rate-reduction objectives (Equation 12) and (Equation 13) are not specifically designed nor engineered for visual quality and the networks and hyper-parameters adopted in our experiments are rather basic compared to many of the state-of-the-art generative methods, our method is still rather competitive in terms of these metrics. In our current implementation, the original objectives are used without any other heuristics or regularization. The simplicity of our framework and formulation suggests that there is significant room for further improvement. For instance, in all experiments on all datasets, we have chosen a feature dimension of d=128 for simplicity and uniformity. In the last [Sec app1dot11-entropy-24-00456], we have conducted an ablation study on using a higher feature dimension d=512. The visual quality of the learned model can be significantly improved (as shown in Figure A22 and Table A14 of [Sec app1dot11-entropy-24-00456]).

In fact, compared to these methods, our method has learned not just any generative model. It has learned a *structured* generative model that has many additional beneficial properties that we now present.

### 3.3. Benefits of the Learned LDR Transcription Model

As we have argued before, the learned LDR transcription model (including the feature z, the encoder *f*, and the decoder *g*) can be used for both generative and discriminative purposes. In particular, unlike almost all existing generative methods, the internal structures or distribution of the learned *z* are no longer “hidden” as they have clear subspace structures. Hence, we can easily derive an explicit (parametrizable) model for the distribution of the learned features as a mixture of independent subspace-like Gaussians. This gives us full control in sampling the learned distribution for generative purposes.

**Principal subspaces and principal components for the feature.** To be more specific, given the learned *k*-class features $\cup_{j=1}^{k}Z_{j}$ for the training data, we have observed that the leading singular subspaces for different classes are all approximately orthogonal to each other: $Z_{i}⊥Z_{j}$ (see Figure 4). This corroborates with our above discussion about the theoretical properties of the rate-reduction objective. They essentially span *k* independent principal subspaces. We can further calculate the mean ${\bar{z}}_{j}$ and the singular vectors ${\left\{v_{j}^{i}\right\}}_{i=1}^{r_{j}}$ (or principal components) of the learned features $Z_{j}$ for each class. Although we conceptually view the support of each class is a subspace, the actual support of the features is close to being on the sphere due to feature (scale) normalization. Hence, it is more precise to find its mean and its support centered around the mean. Here, $r_{j}$ is a rank we may choose to model the dimension of each principal subspace (say, based on a common threshold on the singular values). Hence, we obtain an explicit model for how the feature z is distributed in each of the *k* principal subspaces in the feature space $R^{d}$:(14)$$z_{j}∼{\bar{z}}_{j}+\sum_{l=1}^{r_{j}}n_{l}^{j}\sigma_{j}^{l}v_{j}^{l},\;\mathrm{where}\;n_{l}^{j}∼N\left(0,1\right),\;\;j=1,\ldots ,k.$$

Hence, this essentially gives an explicit mixture of a subspace-like Gaussians model for the learned features: statistical differences between different classes are modeled as *k* independent principal subspaces; statistical differences within each class *j* are modeled as $r_{j}$ independent principal components in the *j*th subspace.

**Decoding samples from the feature distribution.** Using the CIFAR-10 and CelebA datatsets, we visualize images decoded from samples of learned feature subspace. For the CIFAR-10 dataset, for each class *j*, we first compute the top four principal components of the learned features $Z_{j}$ (via SVD). For each class *j*, we then compute $\left|\langle z_{j}^{i},v_{j}^{l}\rangle \right|$, the cosine similarity between the *l*-th principal direction $v_{j}^{l}$ and feature sample $z_{j}^{i}$. After finding the top five $z_{j}^{i}$ according to $\left|\langle z_{j}^{i},v_{j}^{l}\rangle \right|$ for each class *j*, we reconstruct images ${\hat{x}}_{j}^{i}=g\left(z_{j}^{i}\right)$. Each row of Figure 6 is for one principal component. We observe that images in the same row share the same visual attributes; images in different rows differ significantly in visual characteristics such as shape, background, and style. See Figure A7 of [Sec app1dot4-entropy-24-00456] for more visualization of principal components learned for all 10 classes of CIFAR-10. These results clearly demonstrate that the principal components in each subspace of the Gaussian disentangles different visual attributes. In addition, we do not observe any mode dropping for any of the classes, although the dimensions of the classes were not known a priori.

**Disentangled visual attributes as principal components.** For the CelebA dataset, we calculate the principal components of all learned features in the latent space. Figure 7a shows some decoded images along these principal directions. Again, these principal components seem to clearly *disentangle* visual attributes/factors such as wearing a hat, changing hair color, and wearing glasses. More examples can be found in [Sec app1dot6-entropy-24-00456]. The results are consistent with *the property of MCR${}^{2}$ that promotes diversity of the learned features*.

**Linear interpolation between features of two distinct samples.**Figure 7b shows interpolating features between pairs of training image samples of the CeleA dataset, where for two training images $x_{1}$ and $x_{2}$, we reconstruct based on their linearly interpolated feature representations by $\hat{x}=g\left(\alpha f\left(x_{1}\right)+\left(1-\alpha \right)f\left(x_{2}\right)\right),\alpha \in \left[0,1\right]$. The decoded images show continuous morphing from one sample to another in terms of visual characteristics, as opposed to merely a superposition of the two images. Similar interpolation results between two digits in the MNIST dataset can be found in Figure A3 of the [Sec app1dot2-entropy-24-00456].

**Encoded features for classification.** Notice that not only is the learned decoder good for generative purposes, but the encoder is also good for discriminative tasks. In this experiment, we evaluate the discriminativeness of the learned CTRL model by testing how well the encoded features can help classify the images. We use features of the training images to compute the learned subspaces for all classes, then classify features of the test images based on a simple nearest subspace classifier. Many other encoding methods train a classifier (say, with an additional layer) after the learned features. Results in Table 3 show that our model gives competitive classification accuracy on MNIST compared to some of best VAE-based methods. We also tested the classification on CIFAR-10, and the accuracy is currently about 80.7%. As expected, the representation learned with the multi-class objective is very discriminative and good for classification tasks. Be aware that all generative models, GANs, VAEs, and ours, are not specifically engineered for classification tasks. Hence, one should not expect the classification accuracy to compete with supervised-trained classifiers yet. This demonstrates that the learned CTRL model is not only generative but also discriminative.

## 4. Open Theoretical Problems

So far, we have given theoretical intuition and derivation for the formulation of closed-loop transcription, as well as empirical evidence to showcase both the performance and potential of this formulation. In this section, we take a step back to explore the theoretical underpinnings of the closed-loop LDR transcription. We organize this section by discussing three primary objectives associated with learning an LDR representation:*Learn a simple linear discriminative representation f(X) of the data X*, which we can reliably use to classify the data.*Learn a reconstruction g∘f(X)∼X of the so-learned representation f(X)*, to ensure consistency in the representation.*Learn both representation and reconstruction in a closed-loop manner*, using feedback from the encoder *f* and decoder *g* to jointly solve the above two tasks.

These three objectives encompass the overarching principle of CTRL transcription, and indeed each of these objectives are tied to a wide array of mathematical and theoretical problems. We now outline some of the most important theoretical questions or hypotheses implicated by our results, which we leave for future work to study and to answer, likely by a broader range of research communities.

### 4.1. Distributions of the LDR Representation

Our primary mode of optimizing for a “simple representation” is through the LDR framework proposed in [2]. One important open theoretical problem is finding the right energy function to optimize in order to promote LDR. It was shown in [2] that an LDR can be learned for the multi-class data by maximizing the MCR${}^{2}$ objective ΔR(Z) given in (Equation 5). This motivates the first two terms in our objective function (Equation 12): maximizing $\Delta R\left(Z\right),\Delta R\left(\hat{Z}\right)$ promotes their representations to be LDRs.

Although the authors in [2] have shown the MCR${}^{2}$ objective can promote the features learned to be in orthogonal subspaces and characterized the optimal second moments of the distributions, there remain open questions regarding the optimal distributions within the subspaces. A standing hypothesis is that the optimal distributions should be Gaussian. There is indeed already theoretical work on similar energy functions: the Brascamp–Lieb inequalities [67], where the authors study a functional similar to the rate-reduction objective which, in certain contexts, is maximized uniquely by Gaussians. Hence, an important future theoretical direction for the CTRL transcription is to exactly characterize distributional properties of the extremals (both minima and maxima) of the MCR${}^{2}$ objective or its variants. Such results can further justify the use of Gaussian models (Equation 14) to characterize the learned features within the subspaces.

We also notice that the so-learned LDR features have additional striking properties, as shown by examples in Figure 7. Distinctive visual attributes of the imagery data seem to be clearly disentangled by different principal components of the distribution, and along each principal direction, one can linearly interpolate the features, whereas the original data are nonlinear and cannot be directly interpolated. These results go beyond the guarantees given by [2], and an open theoretical problem is that of studying just how the CTRL transcription learns to disentangle and linearize such visual attributes. This understanding is crucial to extend the CTRL transcription framework beyond the 2D vision domain.

### 4.2. Self-Consistency in the Learned Reconstruction

If the learned encoder Z=f(X) is an embedding of the data submanifolds to the subspaces, it should admit an inverse (decoding) mapping $\hat{X}=g\left(Z\right)$. As distributional distance in the data space is hard to come by, the rate reduction $\Delta R\left(Z,\hat{Z}\right)$ gives a well-defined distribution distance between Z and $\hat{Z}$ which is used to enforce similarity between X and $\hat{X}$ in our formulation. Notice that, unlike the KL-divergence or the JS-divergence, the rate reduction is well-defined for degenerate distributions and easily computable in closed-form between mixtures of (degenerate) Gaussians. The third term of Equation (Equation 12), $\sum_{j=1}^{k}\Delta R(Z_{j}\left(\theta \right),{\hat{Z}}_{j}\left(\theta ,\eta \right))$, is exactly this distributional distance, which is minimized only when the estimated second moments of $Z_{j}$ and ${\hat{Z}}_{j}$ are the same. While this distributional distance seems weaker than sample-wise $\ell^{2}$-distance, we observe strong reconstruction performance nevertheless.

Notice that the current objectives (Equation 12) or (Equation 13) do not impose any constraints on the mappings of individual samples. That is, they do not explicitly specify how an individual sample x should be related to its decoded version $\hat{x}=g\left(f\left(x\right)\right)$, or how their corresponding features z and $\hat{z}$ are related. Hence, theoretically, nothing is known about relationships between individual samples and their features. However, somewhat surprisingly, experimental results with the multi-class objective (Equation 12) in next section suggest that they actually can be rather close, at least for the given training samples X. For example, see Figure 5. Of course, one could consider explicitly imposing certain sample-wise requirements in the objectives, such as enforcing $x^{i}$ to be close to ${\hat{x}}^{i}=g\left(f\left(x^{i}\right)\right)$. It has been observed empirically in GANs or VAEs that imposing such sample-wise similarity or dissimilarity would improve visual quality around samples of interest, such as the DC-VAE [42] and the OpenGAN [68]. However, theoretically, how such sample-wise distances or constraints may affect the difficulty or accuracy of learning the correct support and density of the distributions remains an open problem.

### 4.3. Properties of the Closed-Loop Minimax Game

Above are the two primary objectives for CTRL transcription: while the encoder *f* tries to maximize the expressiveness and discriminativeness of the learned LDR representation, the decoder *g* tries to minimize the reconstruction error and coding rates. The competing objectives of the encoder *f* and the decoder *g* naturally lead to a two-player game. In this paper, we have formulated this game as a zero-sum game, namely Equation (Equation 12). Likewise, we have also implemented the most straightforward algorithm for solving this zero-sum game: gradient descent–ascent (GDA) [52], where the minimizer and maximizer take alternating gradient steps. These simplifications into a GDA-optimized zero-sum game were made in order to create a concrete algorithm for our experimentation. However, simplifying to a zero-sum game and GDA is certainly not the only way to solve the more general game described above. This game-theoretic formulation puts CTRL transcription outside of the theoretical realm of [2], since we are no longer finding pure maximizers of ΔR(Z), but rather stable minimax equilibria.

As is the case with GANs, these equilibria may not necessarily be Nash equilibria [50], but rather the more general sense of Stackelberg [69]. So, the problem of studying minimax equilibria of (Equation 12) is likely, in its most general form, quite challenging. Nevertheless, our experiments suggest such equilibria tend to be well-behaved, e.g., having a large range of attraction. Our extensive empirical experiments and ablation studies indicate that, in general, the minimax objective converges rather stably to good equilibria for all the real datasets without any special optimization tricks or particular requirements on the networks. The only important factor for the stability of the optimization seems to be a large enough batch size (see [Sec app1dot10-entropy-24-00456]). These observations can be further corroborated with analysis on simpler models: our ongoing work suggests that if we restrict our attention to simplified data structures (e.g., X distributed on a linear subspace), then one can provide theoretical guarantees that the equilibria become efficiently and correctly solvable by the minimax formulation. Extending such analysis to more sophisticated data structures (multiple subspaces, nonlinear submanifolds) remains an exciting new directions for future research.

Despite many possible pathological solutions to the minimax game, empirically, as we have presented in the previous section (alongside many examples in the Appendix A), the solution found by the simple GDA algorithm generally strikes a good trade-off between expressiveness and parsimony of the learned model. The solution automatically determines the proper dimensions for different classes. Ablation studies in [Sec app1dot10-entropy-24-00456] on the large ImageNet dataset further suggest that this formulation is insensitive to over-parameterization by increasing network width, as long as the batch size grows accordingly. However, a rigorous justification for such good model-selection properties remains widely open.

## 5. Conclusions and Future Work

This work provides a novel formulation for learning a *both generative and discriminative* representation for a multi-class, multi-dimensional, possibly nonlinear, distribution of real-world data. We have provided compelling empirical evidence that the distribution of most datasets can be effectively mapped to an LDR, a union of independent principal subspaces and principal components. The objective function is entirely based on an intrinsic information-theoretic measure, the rate reduction, without any other heuristics or regularizing terms. The objective can be achieved with a closed-loop minimax game between the two encoder and the decoder networks without any additional network(s).

The main purpose of this paper is to demonstrate the conceptual simplicity and practical potential of this new framework for distribution/representation learning, instead of striving for state-of-the-art performance with heavy engineering. Nevertheless, with our preliminary implementation, a more informative LDR of the data can be effectively learned with a simple closed-loop transcription for a variety of real-world, multi-class, multi-modal visual datasets, from small to large, from low-resolution to higher-resolution, from domain-specific to diverse categories. The so-learned encoder *f* already enjoys the benefits of AE/VAEs for their discriminative property and the decoder *g* with the benefits of GANs for their good generative visual quality. However, probably more importantly, the internal structures of the learned feature representation has now become transparent, hence *fully interpretable and controllable* (for generative purposes): visual differences between classes are naturally “disentangled” as independent subspaces, while diverse visual attributes within each class are “disentangled” as principal components within each subspace. From extensive ablation studies given in the Appendix A, we see that the rate-reduction-based objective can be stably optimized across a wide range of datasets and network architectures without any additional regularizations or engineering tricks. Both the *feedback closed-loop* and the *rate-reduction measure* play indispensable roles in fostering the ease and success of finding the CTRL transcription.

One may notice that there are many ways this simple formulation can be significantly improved or extended. Firstly, in this work, we have simply adopted networks that were designed for GANs, but they may not be optimal for the rate-reduction-type objectives. For example, our ablation study already suggests that some of the components of such networks such as spectral normalization are not quite essential. Characteristics from the white-box ReduNet [2] derived from optimizing rate reduction can be explored in the future. Secondly, notice that our rate-reduction objectives do not impose any requirements on how individual samples should be encoded or decoded although the results from the multi-class objective indicate a certain level of alignment on the individual samples. Recent studies such as DC-VAE [42] or OpenGAN [68] suggest that imposing additional regularization on individual samples may further improve decoded visual quality. Such regularization can certainly be incorporated into this new framework. Last but not the least, compared to GANs and VAEs, our method leads to an *explicit* structured model for the feature distribution: a mixture of incoherent subspace Gaussians. Such an explicit model has the potential of making many subsequent tasks easier and better: better control of feature sampling for decoding and synthesis [70], designing more robust generators and classifiers for noise and corruptions based on the low-dimensional structures identified, or even extending to the settings of incremental and online learning [71,72]. We leave all these new directions, together with all the open theoretical problems posed in Section 4, for future investigation.

## Figures and Tables

**Figure 1 entropy-24-00456-f001:**
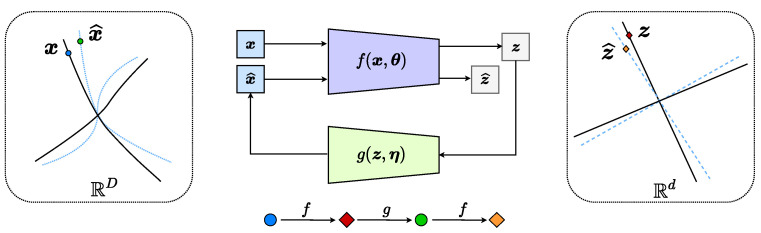
**CTRL: A Closed-loop Transcription to an LDR.** The encoder *f* has dual roles: it learns an LDR z for the data x via maximizing the rate reduction of z and it is also a “feedback sensor” for any discrepancy between the data x and the decoded $\hat{x}$. The decoder *g* also has dual roles: it is a “controller” that corrects the discrepancy between x and $\hat{x}$ and it also aims to minimize the overall coding rate for the learned LDR.

**Figure 2 entropy-24-00456-f002:**
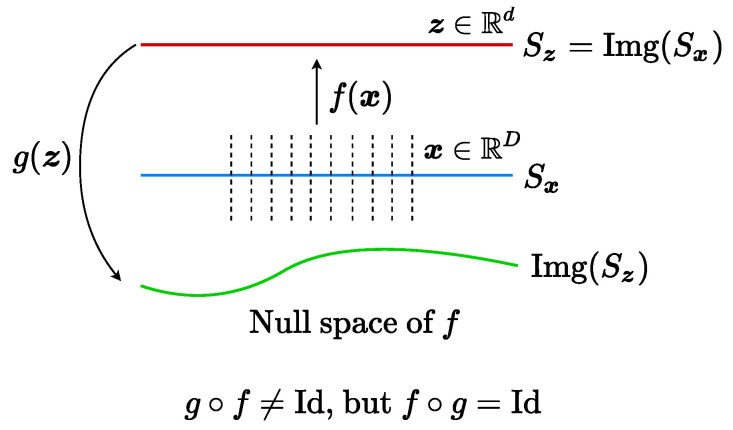
**Embeddings of Low-Dimensional Submanifolds in High-Dimensional Spaces.**$S_{x}$ (blue) is the submanifold for the original data x; $S_{z}$ (red) is the image of $S_{x}$ under the mapping *f*, representing the learned feature z; and the green curve is the image of the feature z under the decoding mapping *g*.

**Figure 3 entropy-24-00456-f003:**
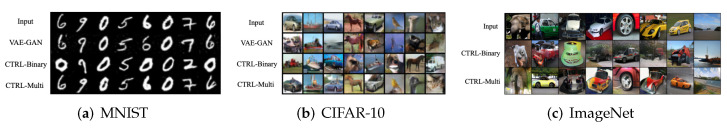
Qualitative comparison on (**a**) MNIST, (**b**) CIFAR-10 and (**c**) ImageNet. First row: original X; other rows: reconstructed $\hat{X}$ for different methods.

**Figure 4 entropy-24-00456-f004:**
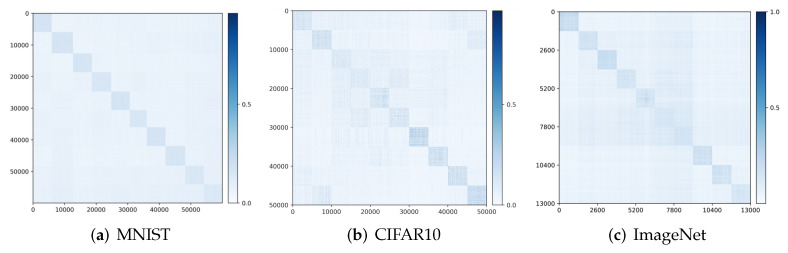
Visualizing the alignment between Z and $\hat{Z}$: $|Z^{⊤}\hat{Z}|$ and in the feature space for (**a**) MNIST, (**b**) CIFAR-10, and (**c**) ImageNet-10-Class.

**Figure 5 entropy-24-00456-f005:**
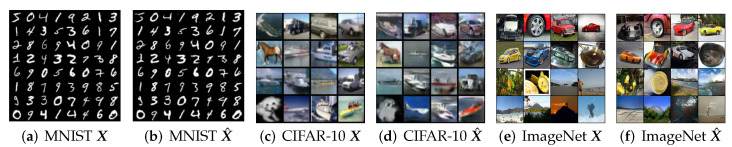
Visualizing the auto-encoding property of the learned closed-loop transcription ($x\approx \hat{x}=g\circ f\left(x\right)$) on MNIST, CIFAR-10, and ImageNet (zoom in for better visualization).

**Figure 6 entropy-24-00456-f006:**
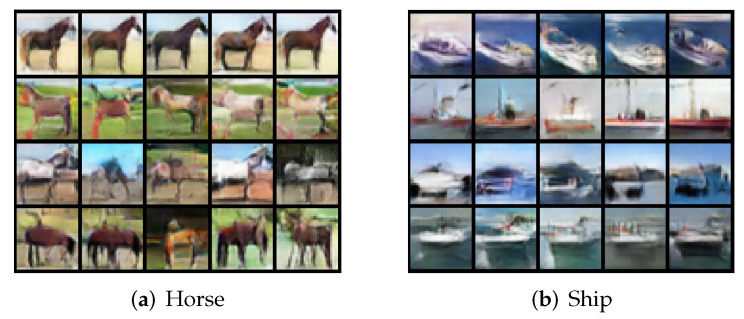
**CIFAR-10 dataset.** Visualization of top 5 reconstructed $\hat{x}=g\left(z\right)$ based on the closest distance of z to each row (top 4) of principal components of data representations for class 7—‘Horse’ and class 8—‘Ship’.

**Figure 7 entropy-24-00456-f007:**
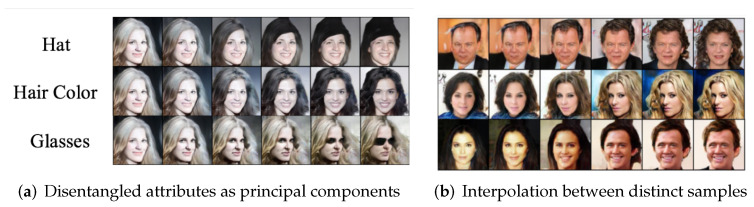
**CelebA dataset.** (**a**): Sampling along three principal components that seem to correspond to different visual attributes; (**b**): Samples decoded by interpolating along the line between features of two distinct samples.

**Table 1 entropy-24-00456-t001:** Quantitative comparison on MNIST and CIFAR-10. Average Inception scores (IS) [65] and FID scores [66]. ↑ means higher is better. ↓ means lower is better.

Method		GAN	GAN (CTRL-Binary)	VAE-GAN	CTRL-Binary	CTRL-Multi
MNIST	IS ↑	2.08	1.95	**2.21**	2.02	2.07
	FID ↓	24.78	20.15	33.65	**16.43**	16.47
CIFAR-10	IS ↑	7.32	7.23	7.11	**8.11**	7.13
	FID ↓	26.06	22.16	43.25	**19.63**	23.91

**Table 2 entropy-24-00456-t002:** Comparison of CIFAR-10 and STL-10. Comparison with more existing methods and on ImageNet can be found in Table A10 in the Appendix A. ↑ means higher is better. ↓ means lower is better.

Method		GAN Based Methods			VAE/GAN-Based Methods				
		SNGAN	CSGAN	LOGAN	VAE-GAN	NVAE	DC-VAE	CTRL-Binary	CTRL-Multi
CIFAR-10	IS ↑	7.4	8.1	**8.7**	7.4	-	**8.2**	**8.1**	7.1
	FID ↓	29.3	19.6	**17.7**	39.8	50.8	**17.9**	**19.6**	23.9
STL-10	IS ↑	**9.1**	-	-	-	-	8.1	8.4	7.7
	FID ↓	40.1	-	-	-	-	41.9	**38.6**	45.7

**Table 3 entropy-24-00456-t003:** Classification accuracy on MNIST compared to classifier-based VAE methods [42]. Most of these VAE-based methods require auxiliary classifiers to boost classification performance.

Method	VAE	Factor VAE	Guide-VAE	DC-VAE	CTRL-Binary	CTRL-Multi
MNIST	97.12%	93.65%	98.51%	98.71%	89.12%	98.30%

## Data Availability

Data and results can be found in Section 3 and Appendix A.

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
