# Peer review of "CTRL: Closed-Loop Transcription to an LDR via Minimaxing Rate Reduction"

_entropy, 2022, doi:10.3390/e24040456_

Round 1
Reviewer 1 Report
The manuscript is devoted to a discussion of the interesting topic of generative and discriminative representation for sets of possibly nonlinear, distribution of real-world data. It is shown that the distribution of most mapped to an union of independent principal subspaces and principal components on the basis of a kind of minimax game between encoder and decoder networks with use of objective function based only on based on the information-theoretic measure, called rate reduction. The provided examples show that the methodology works good. Thus I think that the manuscript will be of interest for the reader society of Entropy. All above determines my positive opinion about the publication of the manuscript.
Author Response
Thank you very much for recognizing the contribution of our work.
Reviewer 2 Report
General Evaluation: it can be accepted after some corrections.
This paper presents an attractive topic in the domain, but the authors need to revise the paper to be accepted in this journal.
Check the paper language and make sure all language errors have been fixed.
The abstract of the paper is not concisely written. The main advantages of the proposed method are not clearly evident from the abstract. The conclusion in the abstract is quite strong against other researchers' work as the experimental results don't support their claim in terms of accuracy or processing time.
Your ideas in the introduction section need to be more comprehensive
Very Long paragraphs are given; revise it. It takes a long time to go through and will mislead the authors
Check the mathematical notation especially for the proposed method
An overview of the related works is needed, which can show future readers more details about the problems and the standard methods that have been used to solve similar problems
When referring related work in the bibliography, adding citations to well-known world-wide journals (especially publications in recent years) would inspire people in its research community take interest in this presentation. For example, the following papers might be cited in your work.
A deep adversarial model for suffix and remaining time prediction of event sequences, Dwarf Mongoose Optimization Algorithm, Ebola Optimization Search Algorithm: A new nature-inspired metaheuristic algorithm.
The discussion is too shallow, did not explain why the proposed method is superior
What are the limitation(s) of the methodology adopted in this work?
The main contribution and originality should be explained in more details
In the conclusion section, it will include research contributions, research limitations, and future works
The references should be updated so as to consider the recent valid publications in this field such as comparisons with methods incorporating deep learning which has been dismissed in the current version.
Author Response
Thank you for the comments. The point-to-point responses can be found in the attached document.

Reviewer 3 Report
- This paper has researched and proposed a new approach to learn a both generative and discriminative representation for various kinds of multi-class multi-dimensional real world datasets, especially the images. The feed-back control idea of the modern control theory is innovatively applied for the modelling and learning for the dataset distribution and representation.
- With the extensive experiments on the widely used benchmark datasets, the new approach is proved to be competitive and generally better than existing methods such as the widely applied GAN, VAE, et al.
- The article is rich in content and information, and is certainly of great value to the relevant researchers.
Some comments and suggestions:
- Similar to the widely used approaches of AE and GAN, it is suggested to define a suitable name for the new approach proposed in this paper, such as LDGR (a both discriminative and generative representation), CLDT (Closed-Loop Data Transcription), LDRT (LDR Transcription), et al. I do not think that both the LDR-Multi and the LDR-Binary are the best name for the new approach.
- Many figures have not been quoted or explained in advance when they are appeared, such as the “Figure 3” is displayed between line 313 and 314 before the first quotation on line 353,which will make the reader more difficult to read.
- Appendix 6 is too long. It is suggested to make some simplifications.
Author Response
Thank you for the comment. The point-to-point responses can be found in the attached document.

Reviewer 4 Report
In this manuscript, the authors make a timely and significant contribution to machine learning by proposing a data transcription method, which models the decoder and encoder in the given particular task as the players of a minimax game. Finding that most dataset’s distributions can be mapped to LDR is noteworthy not only for the present manuscript but also for the white-box ReduNet work. The manuscript is well prepared that besides its length with a rich context, it can be read easily and understood clearly.
Albeit the original “ReduNet” work (reference [2]) is required to understand LDR in detail, the authors have provided sufficient explanation on LDR, so that the manuscript is made self-consistent.
The results are solid and are advancing the field, so that this manuscript can be considered for publication in Entropy.
I would raise small concerns such as revising some sentences to improve the writing of the manuscript, such as the beginning sentences of the Abstract starting with “in particular”, and fixing the typos such as in line 567, “strutured”. I would avoid using some phrases such as “be aware”. Also, the quality of the figures should be improved, because in their present forms, they cannot be understand clearly without zooming in which destroys the viewability due to resolution.
Author Response
Thank you for the reviewer's comments. The point-to-point responses and be found in the attached document.
